Draft Aphaenogaster genomes expand our view of ant genome size variation across climate gradients

http://orcid.org/0000-0003-3758-2406 Lau Matthew K. 1 matthewklau@fas.harvard.edu
http://orcid.org/0000-0003-4151-6081 Ellison Aaron M. 1
http://orcid.org/0000-0002-1378-1606 Nguyen Andrew 2 3
Penick Clint 4
DeMarco Bernice 5
Gotelli Nicholas J. 3
Sanders Nathan J. 6
Dunn Robert R. 7
Helms Cahan Sara 3
1 Harvard Forest, Harvard University , Petersham, MA , USA
2 Department of Entomology and Nematology, University of Florida , Gainesville, FL , USA
3 Department of Biology, University of Vermont , Burlington, VT , USA
4 The Biomimicry Center, Arizona State University , Tempe, AZ , USA
5 Smithsonian Institution , Washington, DC , USA
6 Environmental Program, Rubenstein School of Environment and Natural Resources, University of Vermont , Burlington, VT , USA
7 Department of Applied Ecology, North Carolina State University , Raleigh, NC , USA
Andrew Nigel
Electronic publication date: 2019 Mar 11
Publication date: 2019
Volume: 7
Electronic Location ID: e6447
Received 2018 Apr 25; Accepted 2019 Jan 10
Copyright: © 2019 Lau et al.
Copyright year: 2019
Copyright holder: Lau et al.
License: This is an open access article distributed under the terms of the Creative Commons Attribution License, which permits unrestricted use, distribution, reproduction and adaptation in any medium and for any purpose provided that it is properly attributed. For attribution, the original author(s), title, publication source (PeerJ) and either DOI or URL of the article must be cited.
License URL: https://creativecommons.org/licenses/by/4.0/

Keywords: Ants, Genomics, Ecology, Climate change, Evolution, Adaptation

Funding: US National Science Foundation Dimensions of Biodiversity grant DEB 11-36646 This work was supported by a US National Science Foundation Dimensions of Biodiversity grant (DEB 11-36646) to Nathan J. Sanders, Robert R. Dunn, Aaron M. Ellison, Nicholas J. Gotelli and Sara Helms Cahan. The funders had no role in study design, data collection and analysis, decision to publish, or preparation of the manuscript.

==============================
Given the abundance, broad distribution, and diversity of roles that ants play in many ecosystems, they are an ideal group to serve as ecosystem indicators of climatic change. At present, only a few whole-genome sequences of ants are available (19 of >16,000 species), mostly from tropical and sub-tropical species. To address this limited sampling, we sequenced genomes of temperate-latitude species from the genus Aphaenogaster, a genus with important seed dispersers. In total, we sampled seven colonies of six species: Aphaenogaster ashmeadi, Aphaenogaster floridana, Aphaenogaster fulva, Aphaenogaster miamiana, Aphaenogaster picea, and Aphaenogaster rudis. The geographic ranges of these species collectively span eastern North America from southern Florida to southern Canada, which encompasses a latitudinal gradient in which many climatic variables are changing rapidly. For the six genomes, we assembled an average of 271,039 contigs into 47,337 scaffolds. The Aphaenogaster genomes displayed high levels of completeness with 96.1% to 97.6% of Hymenoptera BUSCOs completely represented, relative to currently sequenced ant genomes which ranged from 88.2% to 98.5%. Additionally, the mean genome size was 370.5 Mb, ranging from 310.3 to 429.7, which is comparable to that of other sequenced ant genomes (212.8–396.0 Mb) and flow cytometry estimates (210.7–690.4 Mb). In an analysis of currently sequenced ant genomes and the new Aphaenogaster sequences, we found that after controlling for both spatial autocorrelation and phylogenetics ant genome size was marginally correlated with sample site climate similarity. Of all examined climate variables, minimum temperature, and annual precipitation had the strongest correlations with genome size, with ants from locations with colder minimum temperatures and higher levels of precipitation having larger genomes. These results suggest that climate extremes could be a selective force acting on ant genomes and point to the need for more extensive sequencing of ant genomes.

Introduction

Understanding how terrestrial ecosystems will respond to ongoing shifts in climatic variables, such as temperature and precipitation, will improve our ability to manage communities and mitigate impacts of climatic change. The mean global temperature is currently on track to meet or exceed that predicted by the most extreme forecasting models (Brown & Caldeira, 2017). Climatic change is also pushing local conditions outside the boundaries of historic ranges, potentially leading to combinations of species or entire ecosystems that have no contemporary analogs (Burrows et al., 2014). As climate-driven impacts on evolutionary responses are likely to occur over contemporary time scales, with the potential for ecological and evolutionary dynamics to affect communities and ecosystem processes (Rowntree, Shuker & Preziosi, 2011; Des Roches et al., 2017), there is a need for a comprehensive study of the genetic basis of species’ responses to climate (Parmesan, 2006).

The biodiversity of most terrestrial systems is great enough to be intractable to study in its entirety. To deal with this, researchers often study “indicator” species whose responses to environmental change are broadly representative of a much wider range of taxa (Siddig et al., 2016). Ants (Formicidae), in particular, are widely used as indicator taxa (Agosti et al., 2000) as they play key roles in community dynamics and ecosystem processes, including key interactions, such as seed dispersal and the movement of soil via colony construction (Del Toro, Ribbons & Pelini, 2012). Ants also are responsive to changes in temperature and other climatic variables via individual responses, changes in colony structure and community assembly (Kaspari et al., 2015; Spicer et al., 2017; Diamond et al., 2017; Diamond & Chick, 2018).

Multiple studies support the perspective that a more complete knowledge of ant genetics will increase our understanding of ant responses to environmental change (Diamond et al., 2012; Nygaard & Wurm, 2015; Stanton-Geddes et al., 2016; Boomsma et al., 2017; Penick et al., 2017). Studies of ant genomes have shed light on the evolution and social organization of ants (Libbrecht et al., 2013). One promising avenue is the possibility of genome size as an adaptive trait in ants. Recent observational studies have reported biogeographic patterns in genome size in arthropod taxa, for example, Crustacea (Hultgren et al., 2018), and patterns in insect genomes suggest that climate may constrain genome size with cold temperatures possibly selecting for larger genome sizes (Mousseau, 1997; Petrov, 2001; Alfsnes, Leinaas & Hessen, 2017). Specific to ants, previous research into genome size variation using flow cytometry found that ants have small genomes relative to other insect taxa and that their genomes display large variation across subfamilies with patterns indicative of both gradual and rapid evolution in genome size (Tsutsui et al., 2008).

At present relatively few ant species have been sequenced—20 in total, of which 19 are currently available in the NCBI Genome Database (accessed 18 December 2018, see Table S1). Of these, most are from tropical and subtropical assemblages (Fig. 1), and all but five represent unique genera (the exceptions being two species of Atta and three of Trachymyrmex). No species of Aphaenogaster, which are abundant ants that play key roles in the dispersal of understory plant species in North America and temperate Asia, have yet been sequenced. Previous studies have also shown that Aphaenogaster species’ ecological and physiological responses to climatic change appear to depend both on species identity and on the geographic region in which climatic change occurs (Warren & Chick, 2013; Stanton-Geddes et al., 2016).

Figure 1 Frequency distribution of currently sequenced ant genomes by geographic location.

Number of ant (Formicidae) whole-genome sequences available in NCBI by country (accessed August 2018).

To increase the number of genomes of temperate-zone ant species, we sequenced the genomes of Aphaenogaster species. We conducted whole genome sequencing for six species: Aphaenogaster ashmeadi, Aphaenogaster floridana, Aphaenogaster fulva, Aphaenogaster miamiana, Aphaenogaster picea, and Aphaenogaster rudis. These species were collected from across a broad biogeographic gradient spanning 10° of longitude and 12° of latitude. We also conducted an initial exploration of biogeographic patterns in ant genome sequences, focusing on genome size. To do this we analyzed the newly collected Aphaenogaster sequences together with all publicly available ant whole genome sequences. We present the newly sequenced Aphaenogaster genomes and investigate biogeographic (i.e., location and climate) related patterns of ant genomes using all currently sequenced ant genomes.

Materials and Methods

Sampling and whole-genome sequencing and assembly

Entire colonies of the six Aphaenogaster species were collected by A. Nguyen and C. Penick from field sites in eastern North America (Fig. 2). Ants were identified to species and voucher specimens have been deposited at the Museum of Comparative Zoology, Harvard University. Individuals from each colony were isolated from nest material and debris, weighed, placed in 50 ml Falcon centrifuge tubes, and immediately flash frozen in a −80 °C freezer. Colony weights were: 794 mg (Aphaenogaster ashmeadi), 652 mg (Aphaenogaster floridana), 520 mg (Aphaenogaster fulva), 749 mg (Aphaenogaster picea), 862 mg (Aphaenogaster miamiana), 280 mg (Aphaenogaster rudis 1), and 236 mg (Aphaenogaster rudis 2).

Figure 2 Sample location map with overlaid photos of sequenced Aphaenogaster species.

We sampled seven colonies representing six species of Aphaenogaster, including (A) A. rudis, (B) A. picea, (C) A. floridana, (D) A. fulva, (E) A. miamiana, and (F) A. ashmeadi from (G) sampling locations across eastern North America (see Table 1). All photos by April Noble (available from http://www.antweb.org).

Table 1 Climate variables for colony sample sites.

Climate are 30 year normal values (1970–2000) for minimum temperature of the coldest month (Tmin), maximum temperature of the warmest month (Tmax), and total precipitation (Precip) from the WorldClim database accessed on 08 August 2018.

	Lat	Lon	Tmin (C)	Tmax (C)	Precip (mm)	
Aphaenogaster ashmeadi	29.785325	−82.031176	5.80	32.70	1,314	
Aphaenogaster floridana	29.785325	−82.031176	5.80	32.70	1,314	
Aphaenogaster fulva	32.692384	−82.514575	1.30	33.30	1,155	
Aphaenogaster miamiana	29.657955	−82.301773	5.90	32.80	1,322	
Aphaenogaster picea	42.6004513	−72.5847494	−12.40	28.30	1,122	
Aphaenogaster rudis1	36.0200847	−78.9830646	−2.70	31.50	1,164	
Aphaenogaster rudis2	36.0200847	−78.9830646	−2.70	31.50	1,164	

Whole colony DNA was used to have sufficient concentrations for sequencing. DNA was then extracted from each colony using methods developed previously for genomic sequencing of whole colonies of colonial mosquitos (Anopheles spp.) (Neafsey et al., 2010) and sequenced using an Illumina HiSeq 2500 at the Broad Institute (Cambridge, MA, USA). A combination of fragment and jump sequences were used to generate higher quality, long sequence reads.

Raw sequences were processed to remove chimeric and contaminant sequences, screened for contaminants by BLAST searches (using blastn) to identify sequences with likely matches to non-target species (primarily Wolbachia and Mycoplasma), and assembled using ALLPATHS-LG (version r48559) (Gnerre et al., 2011). Additional assembly processing using PILON (version 1.13) (Walker et al., 2014) was applied to reduce base-call errors and gaps in coverage. On average, across all seven genomes, PILON reduced coverage gaps by 3.1% or 3.9 Mb. GAEMR (http://www.broadinstitute.org/software/gaemr/) software produced summary statistics of the final assembled genomes. Once assembled, repeat regions in the Aphaenogaster genomes were detected and masked using Repeatmasker (version 4.0.5 Institute for Systems Biology). Genome completeness was assessed for all ant genome assemblies using BUSCO version 3.0.2 (Waterhouse et al., 2018) with the Hymenoptera BUSCO lineage (ODB9 from February 13, 2017) with the honey bee starting genome and HMMER version 3.1b1, BLAST+ version 2.2.31 and AUGUSTUS version 3.0.3 dependencies. For each genome we ran BUSCO on the final contaminant removed, repeat-masked Aphaenogaster assemblies and the most recent published assembly for the NCBI published genomes.

Analysis of genomes along climate gradients

After masking repeat regions, we applied MASH distance (Ondov et al., 2016) to measure pairwise dissimilarity of genomic sequences. The MASH method extends a data compression and dimensionality-reduction algorithm to generate estimates of sequence similarity with low computational overhead. Briefly, the pairs of genomic sequences were pre-processed into sets of k-mers of size 21 with the size of the non-redundant hashes retained set to 1,000. These settings have been demonstrated to provide good representation of genomic similarity with minimal computational costs (Ondov et al., 2016). These sets were then used to estimate the Jaccard similarity coefficient (the ratio of shared k-mers to total k-mers) of subsampled k-mer pairs of genomes. This unbiased estimate of the Jaccard similarity (J) was then used to calculate the dissimilarity of the two genomes (D) as D = 1 − J. All Jaccard similarity estimates had p-values less than 10−14, which is below the recommended 10−3 probability of observing values of J due to chance.

We used multivariate correlation analyses to examine biogeographic patterns of ant genomes. Mantel tests of multivariate correlation of distance matrices were used to examine correlations among ant genomes and climate variables. Specifically, we used directional (H◦: Mantel r ≤ 0) partial mantel tests, which calculate the correlation between two distance matrices while controlling for the covariance with other matrices (Goslee & Urban, 2007). First, we examined the correlations between genomic similarity (MASH distance), whole-genome size similarity (Euclidean distance of assembly size in total base pairs) and climate variables (also using Euclidean distance). Via partial Mantel tests, we were able to isolate the correlation between genome size and climate by controlling for spatial autocorrelation and potential phylogenetic patterns by including geodesic and MASH distances as terms.

We obtained previously sequenced ant whole genome and climate data from publicly available databases. Whole genome sequences for ants were obtained from the NCBI Genome database (accessed August 2018, see Table S1). Climatic variables for each sampling location was obtained from the WorldClim database (version 2.0) at a 2.5 arc minute spatial resolution from the years 1970 to 2000 (Fick & Hijmans, 2017). Although used in the previous analyses of ant genomes, two species, (Wasmannia auropunctata and Monomorium pharaonis), which did not have published location information, were excluded from biogeographic analyses.

Using a permutational multivariate analysis of variance (PerMANOVA) procedure, we parsed the individual variables that were correlated with both genome size and MASH similarity. PerMANOVA is a flexible multivariate analog of ANOVA that permits the use of a wider set of similarity metrics to be used for the response matrix (Anderson, 2001), such as the MASH distance. We ran a total of 10,000 permutations of the original distance matrices for each statistical permutation procedure. We chose a subset of all possible climate variables available via WorldClim for this analysis. A visual inspection of the sampled climate variable correlations indicated that the primary climate variables, mean annual temperature (MAT), minimum temperature of the coldest month (Tmin), maximum temperature of the hottest month (Tmax), annual precipitation (PA) and precipitation seasonality (PS), represented the majority of climate variation (Fig. 3). Based on this, we only included these variables, along with latitude and longitude coordinates, as factors in the PerMANOVAs.

Figure 3 Heatmap of climate variable intercorrelations.

Heatmap of Pearson correlations among climate variables. Cells in the heatmap are colored by the correlation between the two variables that intersect at that location ranging from blue = −1 to white = 0 to pink = 1. The variables are arrayed by hierarchical clustering of the correlations, as shown by the dendrograms on the top and left side. For variable descriptions, see Table 2.

Table 2 WorldClim variables, abbreviations, and numbers for the climate variables used in the analysis of size and MASH similarity of ant genomes.

WorldClim variable	BIO number	
Annual mean temperature (MAT)	BIO1	
Mean diurnal range (MDR)	BIO2	
Isothermality (Iso)	BIO3	
Temperature seasonality (TS)	BIO4	
Max temperature of warmest month (Tmax)	BIO5	
Min temperature of coldest month (Tmin)	BIO6	
Temperature annual range (ATR)	BIO7	
Mean temperature of wettest quarter (MTWeQ)	BIO8	
Mean temperature of driest quarter (MTDQ)	BIO9	
Mean temperature of warmest quarter (MTWaQ)	BIO10	
Mean temperature of coldest quarter (MTCQ)	BIO11	
Annual precipitation (PA)	BIO12	
Precipitation of wettest month (PWM)	BIO13	
Precipitation of driest month (PDM)	BIO14	
Precipitation seasonality (PS)	BIO15	
Precipitation of wettest quarter (PWeQ)	BIO16	
Precipitation of driest quarter (PDQ)	BIO17	
Precipitation of warmest quarter (PWaQ)	BIO18	
Precipitation of coldest quarter (PCQ)	BIO19	

It is important to note that we are using assembly length as an indicator of genome size. As genome size estimates are generally used to set assembly size targets for whole genome sequencing efforts (Hare & Johnston, 2011), we expect there to be a high degree of correlation between assembly size and genome size. Also, as a test of the potential relationship between assembly size and true genome size, we examined the correlation between the average assembly sizes of ant genera that overlapped with flow cytometry estimates of those published in Tsutsui et al. (2008). We applied leave-one-out outlier detection (Cook & Weisberg, 1982) to least squares regression analysis examining the predictive potential of assembly length and flow cytometry size estimates at the genus level and found one strong outlier (Fig. S1), Dinoponera quadriceps, whose assembly size was less than half the flow cytometry estimate (259.7 vs. 554.7 Mb). After removing this species from the regression analysis we found a strong, significant relationship between assembly length and genome size (R2 = 0.76, F1,5 = 15.99, p-value = 0.010), which supports the use of assembly length as an indicator of genome size. Based on this, we did not include D. quadriceps in any of the analyses examining patterns of assembly length.

Data, computation and statistics

The raw and assembled genome sequences are currently archived at Harvard Forest (Petersham, MA, USA) and in NCBI’s genome database (Genome accessions NJRK00000000–NJRQ00000000 and BioSample accessions SAMN06894590–SAMN06894596). Genomic distance (MASH) computations were run on the Odyssey cluster supported by the FAS Division of Science, Research Computing Group at Harvard University. All analyses were conducted in R Core Team (2017). Analytical scripts for the project have been versioned and archived (DOI: 10.5281/zenodo.1341982) and are available online at https://zenodo.org/record/1341982. We used the vegan (Oksanen et al., 2016) and ecodist (Goslee & Urban, 2007) packages in R for multivariate analyses.

Results

Genome quality and composition

DNA extractions yielded substantial amounts of high quality DNA with concentrations and quality scores ranging from 3.45–5.39 to 4.05–4.27 ngμL−1, respectively. All genome assemblies displayed good coverage, with an average of 70% across all genomes (Table 3). Across all species, the length of the shortest contig at 50% of the genome (i.e., N50) was 18,864 bases; average assembly GC content was 38.18%; and average genome size was 370.45 Mb. Using GAEMR’s BLAST feature to conduct a search of the contigs against the NCBI’s nucleotide sequence database, we discovered that 38.98% and 22.04% of the top hits were “ant” and Aphaenogaster, respectively. Using the Hymenoptera BUSCO set, the Aphaenogaster genomes displayed a high degree of completeness. Between 96.1% and 97.6% with an average of 97.0% ± 0.2 (S.E.) of Hymenoptera BUSCOs were completely represented. The Aphaenogaster genomes were among the more complete genomes, relative to the other sequenced ant genomes from NCBI, which ranged from 88.2% to 98.5% complete Hymenoptera BUSCOs with a an average of 96.3% ± 0.5 (S.E.) (Fig. 4A). Also, the sizes of the Aphaenogaster genomes were within the range of other ant genomes based on size from both flow cytometry (Tsutsui et al., 2008) and the previously sequenced ant genomes available in NCBI (Fig. 4B).

Table 3 Sequencing statistics for the genomes of the sequenced colonies of Aphaenogaster.

	A. ashmeadi	A. floridana	A. fulva	A. miamiana	A. picea	A. rudis1	A. rudis2	
Total scaffold length (Mb)	310.33	382.86	346.13	342.64	386.04	395.41	429.70	
Coverage (%)	81.46	71.88	70.70	77.40	67.47	66.49	65.59	
Scaffold N50 (bp)	336,807.00	439,114.00	255,328.00	351,517.00	322,984.00	300,103.00	269,776.00	
Scaffolds	5,087.00	6,422.00	7,031.00	6,920.00	6,808.00	7,404.00	7,665.00	
Max gap (bp)	13,070.00	15,108.00	12,104.00	11,453.00	14,952.00	18,586.00	24,564.00	
Captured gaps	26,350.00	30,858.00	32,881.00	28,801.00	36,417.00	34,062.00	34,313.00	
Total gap length (Mb)	57.69	107.89	101.40	77.64	125.15	131.71	148.75	
Total contig length (Mb)	252.64	274.96	244.73	265.00	260.90	263.70	280.95	
Contig N50 (bp)	21,677.00	23,448.00	15,753.00	20,738.00	15,440.00	15,622.00	18,941.00	
Contigs	31,437.00	37,280.00	39,912.00	35,721.00	43,225.00	41,466.00	41,978.00	
Assembly GC (%)	38.27	38.03	38.39	38.21	38.32	38.25	37.88	
Contaminants (%)	0.30	0.24	0.02	0.26	1.14	1.25	0.61	

Figure 4 Frequency distribution of sequenced assembly and flow cytometry estimated genome sizes with overlaid Aphaenogaster genome sizes.

The Aphaenogaster genomes displayed a high degree of completeness and were within the size range of previously published ant genomes. (A) Stacked bar plot of the BUSCO metrics including the number and percentage of complete Hymenoptera (ODB9) BUSCOs (C), also including both single (S) and duplicated (D) copies, as well as fragmented (F) and missing (M) out of 4,415 BUSCOs for the Aphaenogaster and previously published ant genomes available from NCBI (accessed August 2018). (B) Frequency distribution of previously published genome size estimates using flow cytometry from Tsutsui et al. (2008) and those available in the NCBI Genomes database. Vertical lines identify the sizes of the Aphaenogaster assemblies (see Table 3).

Using the MASH genomic distances, we observed patterns of genomic similarity that were in line with expectations of ant relatedness. Sequences formed groups that corresponded with subfamilies (Fig. 5). Aphaenogaster clustered with other genera from the Myrmicinae and, in general, subfamily level clustering tended to follow previously observed patterns of subfamily relatedness (Bolton, 2006; Moreau et al., 2006; Ward, 2014). The Aphaenogaster sequences formed a single cluster containing only Aphaenogaster species and displayed intra-generic levels of genomic variance comparable to other genera (e.g., Trachymyrmex spp.). The separation of the two Aphaenogaster rudis species was initially surprising, as these two samples were collected at the same site (Duke Forest, Durham, NC, USA) and were identified as the same species based on their morphological characteristics (Ellison, 2012; DeMarco & Cognato, 2016). However, two recent studies of targeted gene regions have demonstrated the polyphyletic nature of Aphaenogaster rudis. One study of the evolution of the subfamily Myrmicinae observed that the genus as a whole could be split into at least four different lineages (Ward et al., 2015). Another, more detailed study of the genus in North America found that multiple individuals of Aphaenogaster rudis separated out into distinct groupings, each with other species, specifically, individuals of Aphaenogaster rudis from North Carolina (USA) were observed to form distinct clusters with individuals of Aphaenogaster carolinensis, Aphaenogaster miamiana, Aphaenogaster lamellidens and Aphaenogaster texana (DeMarco & Cognato, 2016).

Figure 5 Heatmap of MASH genomic distances for all sequenced ant genomes.

Heatmap of the MASH genomic distances of the Aphaenogaster species that we sampled together with other ant species in NCBIs. Heat colors shown in the central matrix range from high (white = 1) through moderate (orange = 0.5) to low (red = 0) genomic distance; the diagonal is entirely red because it illustrates the distance of each sequence to itself. The cladograms on the left and top show hierarchical clustering of the genomes. Colors shown to the left of the matrix indicate ant subfamilies: Ponerinae (dark blue), Formicinae (green), Pseudomyrmecinae (pink), Dolichoderinae (red), Dorylinae (yellow), Myrmicinae (light blue).

Biogeographic patterns of ant genomes

After controlling for both spatial autocorrelation and potential phylogenetic patterns, we found a marginally significant, positive correlation between ant assembly size similarity and climate similarity (Mantel R = 0.12, p-value = 0.068). Although assembly size similarity and MASH genome similarity were not significantly correlated (p-value = 0.126), we included MASH as a covariate in addition to geodesic distance because previous research indicated that genome size is associated with phylogenetic relatedness (Alfsnes, Leinaas & Hessen, 2017). We found that different spatial and climatic variables were associated with the size similarity of ant genomes. Longitude but not latitude was a significant predictor of assembly size similarity (Table 4). Temperature of the coldest (Tmin) and hottest (Tmax) month and total PA, were significant predictors, but neither MAT nor PS were significant predictors of assembly size similarity. Overall, Tmin and PA had the strongest relationships with assembly size similarity with both exhibiting generally negative relationships with size (Fig. 6). When the newly sequenced Aphaenogaster genomes were excluded from the analysis, only PA was a significant predictor of assembly size similarity (R2 = 0.31, p-value = 0.020 and see Table S2).

Table 4 PerMANOVA pseudo-F table for the analysis of the relationship between climate variables and ant assembly size similarity.

	df	SS	MS	Pseudo-F	R2	p-value	
Assembly size similarity	
Lat	1	2,428.30	2,428.30	1.70	0.03	0.2093	
Lon	1	9,528.44	9,528.44	6.67	0.11	0.0165	
MAT	1	147.67	147.67	0.10	0.00	0.7531	
Tmin	1	18,869.62	18,869.62	13.22	0.22	0.0029	
Tmax	1	8,994.04	8,994.04	6.30	0.11	0.0230	
PA	1	22,424.69	22,424.69	15.71	0.26	0.0011	
PS	1	1,062.82	1,062.82	0.74	0.01	0.4132	
Residuals	15	21,414.46	1,427.63		0.25		
Total	22	84,870.04			1.00		

Figure 6 Bivariate plot of the correlation between sampling location minimum temperature of the coldest month (Tmin), annual precipitation (PA), and genome size.

Bivariate plot showing the relationship between ant assembly size and minimum temperature of the coldest month (Tmin) with Annual Precipitation (PA) overlaid as colors. Ants from locations with lower minimum temperatures and annual precipitation tended to have larger genomes.

Discussion

We have produced seven draft whole-genome sequences of six species of ants in the genus Aphaenogaster. The addition of the Aphaenogaster sequences increases the breadth of global ant genomic sampling, as these are the first whole-genomes from a previously un-sequenced genus, adding to the sequences of the diverse “formicoid” clade, which contains 90% of all extant ant species (Ward, 2014). Our genomic sequences were comparable in quality to other ant and insect genomes and the patterns of genomic similarity were in line with expectations based on current ant systematics. With the addition of the new Aphaenogaster sequences, our initial biogeographic analysis revealed that ant genomes from more similar climates have more similarly sized genomes with minimum temperatures having the strongest correlation with genome size, which is consistent with the hypothesis that climate has been a force shaping ant genome size.

Although correlative, our genome analysis results are consistent with the hypothesis that ants from regions with more similar climates tend to have similar sized genomes. Previous studies have observed physiological and ecological responses of ants to climate gradients and shifting temperatures (Warren & Chick, 2013; Stanton-Geddes et al., 2016; Diamond et al., 2016, 2017; Nguyen et al., 2017; Helms Cahan et al., 2017; Penick et al., 2017) that could act as agents of selection or as environmental filters. For example, Warren & Chick (2013) found that cold, but not warm, temperatures limited shifts in the distributions of Aphaenogaster picea and Aphaenogaster rudis. Diamond et al., (2016) reported that the rate of colonization and occupancy of nests by Aphaenogaster species in a 5-year experimental warming study (Pelini et al., 2014) declined with temperature in the warm, southern study site (Duke Forest, Durham, NC, USA) but not in the cooler, northern study site (Harvard Forest, Petersham, MA, USA). In addition to the direct impacts of climate, some studies support the importance for the indirect effects of climate via biotic interactions. For example, the distribution of the species Atta texana is limited by the cold-tolerance of its fungal symbiont, cultivars of the genus Attamyces (Mueller et al., 2011). The evolution of the ant-fungus relationship has led to reductions in some ant species ranges by cold temperatures.

Although we found support for increasing genome size with colder temperatures and PA, we are cautious to offer possible mechanisms for these patterns. In addition to these variables themselves being correlated, in general arthropod genome size appears to be influenced by a complex array of selection pressures. This is evidenced by the recent study by Alfsnes, Leinaas & Hessen (2017), which found that genome size patterns varied greatly among major arthropod taxa with high potential for different mechanisms affecting genome size. For example, insects displayed clear phylogenetic correlations with genome size while genome size patterns in crustaceans were nearly independent of phylogeny but strongly related to biogeographic gradients (e.g., decreasing genome size with increasing maximum observed latitude). In addition, Hultgren et al. (2018) found evidence for increasing genome size with latitude in crustaceans but not decapods, adding another example of the potential complexity of genome size as an adaptive trait.

There is the potential for both direct and indirect selection for increased genome size in colder conditions. Hessen et al. (2010) has proposed that there is a complex set of relationships among genome size, developmental rate, cell size, and body size and that selection can act at different points in this causal network. One possible direct pathway is that increased expression of gene products that deal with cold stress could lead to larger genomes via whole genome or individual gene duplications (Dufresne & Jeffery, 2011). Previous work with Aphaenogaster supports this hypothesis, as Stanton-Geddes et al. (2016) found that exposure to extreme cold induced expression of genes in the cold-climate Aphaenogaster picea more so than in Aphaenogaster carolinensis (a more southern, warm climate species). An example of a possible indirect pathway is that cold could select for increased body size to deal with heat-loss in cold conditions (Brown et al., 2004), which could lead to larger genome sizes. There is evidence of cold selecting for greater body size (i.e., Bergmann’s Rule) in ants (Heinze et al., 2003; Bernadou et al., 2016), and some have hypothesized that increased body size could lead indirectly to increased genome size via increased cell size (Ryan Gregory, 2005); however, the most recent, broad analysis of genome size in ants (that we are aware of) did not find support for a relationship between ant genome size and body size after controlling for phylogenetic patterns (Tsutsui et al., 2008). Therefore, assuming that our analysis adequately controlled for phylogenetics, indirect selection on genome size via body size is not a likely explanation for our observed relationship between genome size and temperature.

It is important to keep in mind that the climate related genomic patterns observed in this study should be considered an initial view of possible biogeographic patterns in ant genomes. As the addition of the these sequences had a marked impact on the statistical results of the climate analysis (see Table S2), we expect that further sequencing work will continue to enhance our understanding of the ecological genomics of ants. Also, these findings should be tested with additional sequencing efforts, as we could not control for several potentially important intercorrelated variables. Factors such as sampling bias and sequencing methodology (e.g., 454 vs Illumina) also varied among sequencing efforts, which could have contributed to some of the observed correlations with climate. We did not attempt to control for these factors statistically due to the limitations of the current ant genome sample size. Future work should methodologically and/or statistically control for such sources of variation in ant genomes as more sequences become available to elucidate clearer patterns and resolve underlying ecological and evolutionary mechanisms.

Conclusion

Although we have increased the total number of sequenced ant genomes by over 30%, the total number of ant sequences analyzed here is still a relatively small sample (n = 26) of the estimated >16,000 ant species and subspecies (www.antweb.org, accessed 20 August 2018). Efforts such as the global ant genomics alliance (Boomsma et al., 2017), which aims to greatly increase the number of ant species sequenced from across the world, will provide additional resources for ecological genomics studies. Further work investigating the variation in genomic content and mapping of target coding regions from previous physiological (Nguyen et al., 2017), biochemical (Helms Cahan et al., 2017), and transcriptomic (Stanton-Geddes et al., 2016) studies of Aphaenogaster and other ant species will inform predictions of how these species, and the ecosystems that they inhabit, may respond to ongoing climatic change. For instance, determining the genomic factors underlying the temperature response of ant assemblages to climatic gradients (Warren & Chick, 2013; Diamond et al., 2016, 2017) could provide useful insights into the response of these important organisms to non-analog ecosystem states and idiosyncratic community responses (Bewick et al., 2014). In addition, as species distribution models have been significantly improved by the inclusion of genetic information (Ikeda et al., 2016), an ecological genetics approach that couples ant genomic and ecologically relevant data will provide a useful window into the response of many terrestrial ecosystems to a changing climate.

Supplemental Information

Supplemental Information 1 NCBI genome database accession information (TEX file).

NCBI genome database accession information for the previously sequenced ant genomes and coordinates for species those species that could be obtained from the published literature.

Click here for additional data file.

Supplemental Information 2 NCBI genome database accession information (PDF file).

NCBI genome database accession information for the previously sequenced ant genomes and coordinates for species those species that could be obtained from the published literature.

Click here for additional data file.

Supplemental Information 3 PerMANOVA pseudo-F table for the analysis of the factors correlated with ant genome size (TEX file).

PerMANOVA pseudo-F table for the analysis of the factors correlated with ant genome size only including the previously sequenced NCBI ant specimens.

Click here for additional data file.

Supplemental Information 4 PerMANOVA pseudo-F table for the analysis of the factors correlated with ant genome size (PDF file).

PerMANOVA pseudo-F table for the analysis of the factors correlated with ant genome size only including the previously sequenced NCBI ant specimens.

Click here for additional data file.

Supplemental Information 5 Outlier detection of assembly size of Dinoponera quadriceps.

Outlier detection showing that Dinoponera quadriceps falls above the threshold for outliers, in this case 4 times the mean Cook’s distance from the regression of assembly length and genome size based on flow cytometry.

Click here for additional data file.

Thank you to the genome sequencing team at the Broad Institute (particularly, James Bochicchio, Sarah Young, Terrance Shay and Caroline Cusick) and to Manisha Patel at Harvard Forest’s Torrey Lab for assistance in obtaining and properly storing ant colonies.

Additional Information and Declarations

Competing Interests

Author Contributions

DNA Deposition

Data Availability

Aaron M. Ellison is an Academic Editor for PeerJ.

Matthew K. Lau conceived and designed the experiments, performed the experiments, analyzed the data, prepared figures and/or tables, authored or reviewed drafts of the paper, approved the final draft.

Aaron M. Ellison conceived and designed the experiments, contributed reagents/materials/analysis tools, authored or reviewed drafts of the paper, approved the final draft.

Andrew Nguyen conceived and designed the experiments, performed the experiments, contributed reagents/materials/analysis tools, authored or reviewed drafts of the paper, approved the final draft.

Clint Penick performed the experiments, contributed reagents/materials/analysis tools, authored or reviewed drafts of the paper, approved the final draft.

Bernice DeMarco performed the experiments, authored or reviewed drafts of the paper, approved the final draft.

Nicholas J. Gotelli authored or reviewed drafts of the paper, approved the final draft.

Nathan J. Sanders conceived and designed the experiments, authored or reviewed drafts of the paper, approved the final draft.

Robert R. Dunn conceived and designed the experiments, authored or reviewed drafts of the paper, approved the final draft.

Sara Helms Cahan conceived and designed the experiments, contributed reagents/materials/analysis tools, authored or reviewed drafts of the paper, approved the final draft.

The following information was supplied regarding the deposition of DNA sequences:

The assembled Aphaenogaster whole genome sequences have been deposited in NCBI’s Genomes Database with BioSample accessions (https://www.ncbi.nlm.nih.gov/bioproject?LinkName=biosample_bioproject&from_uid=6894596) SAMN06894590–SAMN06894596 and Genome accessions NJRQ00000000–NJRK00000000.

The following information was supplied regarding data availability:

Analytical code is available at https://github.com/HarvardForest/apGenomes

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
