# Peer review of "Draft Aphaenogaster genomes expand our view of ant genome size variation across climate gradients"

_PeerJ, doi:10.7717/peerj.6447_

## Round 0.1 · original submission · Major Revisions

All three reviewers have provided substantive and insightful comments. Please assess and review accordingly. Address each point from each of the three reviewers in your rebuttal.

Reviewer 1 ·

Basic reporting

The article is generally well prepared, though see comments below regarding climate variables and the abbreviations. Figure are generally crisp / easy to read (save vectors in Figure 6).

A few areas that require attention include:

(1) I think the manuscript could be improved through the addition of more background / context regarding the relationship(s) between climate adaptation, genome size, and response to climatic change.

Given the exploratory nature of the study, perhaps it is not possible to develop predictions / testable hypotheses, but it was unclear to me how / why relationships between genome size and climate gradients would be indicative of ant responses to climate change. In think this is especially important given that the authors were apparently not able to control for phylogenetic relatedness (which they acknowledge and discuss).

The title makes a link between 'ecological genomics' and 'responses to climate change', but that's not really what this paper is doing. Instead, the study involves correlating aspects of the genome with climate variables, finding some significant relationships, and then claiming that these relationships are somehow informative (or at least warrant consideration) of climate change responses. However, without a clear, mechanistic linkage between climate, genome size, and climate adaptation, I could not follow how these biogeographical relationships say anything about climate change responses.

I think it is enough to report these relationships and speculate some in the Discussion as the what they might mean, but as written, I do not feel the title reflects the study or its findings.

(2) At times the writing was vague and wandering, particularly in the Introduction. A good example is L69-70, which states "...phylogenetics is a factor determining the response of ant species to climatic change...". It was unclear to this reader how / why "phylogenetics" would "determine" a response to climatic change.

The Introduction also reads a bit like a laundry list of reasons justifying what it is important to sequences the genomes of ants and Aphaenogaster in particular. In my opinion, a lot of this wasted space. I think it is enough to say ants are important generally, Aphaenogaster is something the authors have studied and so have some insights and get on with a discussion about genomics and climate adaptation - either broadly or in ants specifically to the extent that is possible.

(3) A few minor points:

L168-170 - Check grammar / missing word.

Figure 1 - Not sure this one is necessary.

Figure 6 (vectors and associated labels need to be much darker to be visible).

There is some minor confusion regarding which climate variables / datasets were used when. For example-

Table 2 - PRISM data are reported here, but worldclim is used in the analyses - any reason for different datasets?

L178 - Annual minimum temperature and annual maximum temperature are not among the 19 standard bioclim variables from WordClim - are these custom or referring to BIO5/6 (which are month not annual)?

Experimental design

As mentioned above, I think the manuscript would benefit if predictions or research questions were more clearly articulated and supported in the Introduction. While two hypotheses are mentioned in the final paragraph of the Introduction, they tended to be vague or poorly worded.

For example: "...to test the hypothesis that climate variables shape the distribution of ant genomes, we explored the correlation between spatial and multi-decadal climate variables". First, what is the distribution of an ant genome? And second, it reads as if the correlation of interest is between "spatial and multi-decadal climate variables" - in other words, between two kinds of climate variables, not between a set of climate variables and some aspect of ant genomes.

Otherwise, the methods are well described and the statistical analyses seem well executed.

Validity of the findings

I will echo my comments above regarding the failure to clearly articulate the linkages between climate gradients, genome attributes (size, similarity), and responses to climate change.

While the data seem robust (I cannot comment on the sequencing / bioinformatics) and the analyses sound, and while the authors acknowledge the importance of other factors ignored (for now) in the analyses (such as phylogenetic correlations), I do feel that, given the sample size, the data and results are somewhat limited in what they can tell use about climate adaptation / responses in this group of ants. It certainly is great to have and report these new sequences, which will add to a growing body of sequence data for ants, I just caution against over-interpretation / speculation given the small sample size and uncertainty.

·

Basic reporting

no comment

Experimental design

no comment

Validity of the findings

no comment

Additional comments

Title: Expanded view of the ecological genomics of ant responses to climate change

The authors have produced draft sequences for the genomes of six species of ants in the genus Aphaenogaster collected from eastern North America. There is high value in these ant genome sequences: ants in this genus are models for studies of ecology and behavior (>300 papers with Aphaenogaster in the title), no sequence from this genus had to date been reported, these six new genomes increase the number of ants with sequenced genomes by ~25%.

The methods for genome sequencing and analysis are sound and the writing style and quality are appropriate (see minor issues below). The new ant genome sequences could represent a publication on their own, even without much analysis. The authors perform superficial comparisons of the genome sequences: showing higher similarities between genomes of species in the same genus than between genuses, as well distinct separation from genera compared with others previously published ant genomes in the formicoid subfamily. Intriguingly, the data also support the polyphyly of A. rudis as described in recent literature, though this evidence is based on a sample size of n=2. Together, these data would be sufficient for publicaiton in peerj.

However, the manuscript is framed within the context of using ants as indicator taxa for environmental disturbance and climatic variation. Specifically, the authors suggest that ecological variables should influence ant genomic signatures. They propose that (i) ants found within similar ecological conditions should have more similar genomes; and that (ii) genome size should similarly correspond to biogeographic patterns. These hypotheses are original and would be very interesting if proven (stronger datasets to test such hypotheses may exist in mammals). Unfortunately, neither hypothesis is very well substantiated and the supporting data is rather tenuous because the authors (i) lack power (a small number of species for a relatively high number of examined environmental variables) and (ii) do not control for phylogeny. We note that additionally there is little consideration of possible variation within species.

Furthermore, the title evokes responses to climate change - which implicitly suggests temperature manipulation experiments - but this is quite far from the paper.

This is somewhat frustrating given that the senior author has a strong track record of rigorous work. Controlling for phylogeny/ would make the climatic/biogeographical analysis much stronger, although we note that 21 of the 25 ant species used come from the Americas thus potentially creating a bias. A rigorous analysis of the proposed hypotheses would likely require much additional data generation that is likely beyond the scope of this paper.

We recommend that the authors:
- Move the biogeographic/climatic analysis from results to discussion. And shorten it substantially in the main text (this could involve moving some aspects to supplementary).
- Make it clear that the manuscript mainly reports a powerful new phylogenomic resource for an important ant genus
- Change the title and abstract (and other parts) of the manuscript to focus on the resource and dramatically tone down interpretations linked to ecology or effects of climate change.

We expect that this would only represent a small amount of work.

With kind regards,
Gino Brignoli & Yannick Wurm

Minor comments
L52 “...social structure and community assembly” - missing u
L89 “With the these new…” - remove “the”
L94: "demonstrated patterns in the evolutionary dynamics of ant genome size (Tsutsui et al., 2008) " - I think the authors simply mean to say that there was variation in genome size.
L113: Anophales -> Anopheles
L118 "blast" is lowercase - while subsequent page it is uppercase L128
L118: which version of BLAST? What was approach used to remove contaminants, chimeric sequences etc (software, parameters etc)
L126: what does "70% of fragments mapped " mean? Fragments of what? Mapped from where to where?
L128: using -> Using
L128/129: which NCBI sequence database? NR? Which blast algorithm? Blastx? blastn?
L130: the -> The. similar capitalization issues in Fig 3 legend and elsewehre
L133: I don't understand what "recommended for gene coverage" means?
L123: "Genome Quality:" -> usually the CEGMA (or BUSCO) metrics are used as biological measures of genome completeness.
L189 to 191 - were all 26 ant species used for all analyses? If so which collection locations were used for previously published genomes? (Such information may unfortunately be absent from previously published genomes)
Pg13 “Figure 6. Plot an showing…” - remove “an”
L249 “from from previous experimental…”
Table 1 can be supplementary
Ref "The evolution of genome size in ants. " - should only have 4 authors but has about 20x more!
The two rudis colonies do not cluster together according to whole-genome MASH. I understand the authors dont want to do fullgenome comparisons here - however does e.g., a mitochondrial or nuclear “housekeeping” gene phylogeny based on sequences from these sample support the polyphyly?

Reviewer 3 ·

Basic reporting

no comment

Experimental design

Major comments:
There are a few issues with the main analysis correlating environmental variables with genomic characteristics:
1. The total length of assemblies is not a good metric for genome size. While flow cytometry is ideal, k-mer analysis (e.g. GenomeScope) also tends to be more accurate than assembly size so should be used instead.
2. The authors note that they did not perform phylogenetic correction for their analyses (line 237) but seem to acknowledge that it would be a more appropriate approach. I would argue that such a correction is essential. The conclusion that climate influences genomic characteristics is completely dependent on this analysis.
3. It appears that the latitude and longitude of collecting site were used for correlational analyses. However, given that this study is comparative in nature, each individual genome is meant to represent an entire species. And several of these species have very large ranges that span the entire eastern United States into Canada making these data points inappropriate for this analysis. The midpoint of the species ranges could be a reasonable alternative. Unless the authors are arguing that these six specimens that they have used are effectively a single population but I don’t believe that this is the case. While there appears to be some undiscovered species diversity within A. rudis, the rest appear to be good species without gene flow between them.
4. The previously sequenced ant genomes included in the analysis were sequenced using a variety of different methods. For example, some genomes included 454 sequencing. This could bias the size of the genome assembled. Perhaps some of these influences could be taken into account.
5. The authors perform their analyses both by including Aphaenogaster genomes and again without these new genomes. However, a third analysis that focuses just on Aphaenogaster might be more compelling. This would eliminate some of the phylogenetic bias in the data and also eliminates many of the other factors that could be playing a role as all Aphaenogaster genomes were collected and assembled in the same way. I also wonder if leaving out the fungus-growing ants, a group that is also over-represented, could have a similar effect. Relatedly, the fact that Aphaenogaster has such an impact on the analyses seems to suggest that phylogeny is playing a significant role.

Other comments:
-More detail is needed for the DNA extraction procedure. Was it really necessary to collect DNA from a whole colony? Is there polygyny or multiple-mating in these species that could affect the assembly? Reporting heterozygosity/genetic diversity would be useful and these data could be used to infer colony structure if it is not already known.
-While genome annotation is certainly a significant additional challenge, comparing gene sets across species could provide additional insight. This can now be relatively easily accomplished with pipelines like BRAKER which don’t necessarily require transcript data to run.
-Line 128: That is not the average genome size of those listed in the table
-Line 128-130: Why was this BLAST analysis done?

Validity of the findings

no comment

Additional comments

This study by Lau and colleagues presents the newly sequenced genomes of six species of ants in the genus Aphaenogaster. Given their worldwide abundance and ecological importance, the genomes of these species are likely to be of substantial scientific use. There are relatively few taxonomic groups, particularly within ants, where multiple closely related species have been de novo sequenced, making this a potentially valuable resource for studying both ants and genome evolution.

---

## Round 0.2 · Minor Revisions

Review 2 has some pressing issues that need to be addressed fully and considered in your revised manuscript and rebuttal letter.

Reviewer 1 ·

Basic reporting

I feel the authors have fully addressed all concerns raised during the previous round of review.

Experimental design

No comment.

Validity of the findings

I feel the authors have fully addressed all concerns raised during the previous round of review.

Additional comments

Thank you for your careful attention to concerns raised during the first review. I feel the paper is much improved and will make an excellent early contribution to our understanding of any genomes.

·

Basic reporting

see below

Experimental design

see below

Validity of the findings

see below

Additional comments

Line 242 p=0.076 is not "marginally significant"


As reviewer 3 points out, genome assembly size is indeed very noisy and something like GenomeScope is much more powerful and precise for this type of comparison. The response to the reviewers comment is inapporpriate. You cannot have a genome assembly if you didn't start with short reads. This is a very very quick analysis (half a day MAX to do all species) but would increase accuracy of these estimates.



### My additional comments in response to the cover letter ###

L123: "Genome Quality:" -> usually the CEGMA (or BUSCO) metrics are
used as biological measures of genome completeness.
RESPONSE: CEGMA has been discontinued and the authors recommended
using alternative tools. BUSCO is available, however, at present we
have not completed the genome annotation process, which appears to be
a requirement.

### What is this statement based on?
### Having an existing geneset is NOT a requirement for running BUSCO.




Ref "The evolution of genome size in ants. " - should only have 4
authors but has about 20x more!
RESPONSE: We have not changed this, as we are using the bibtex cls
file that was provided. Please let us know if there is a different
formatting file that we should use.

### I do not understand what the authors are saying.
### The manuscript mentioned clearly has a specific set of authors.
### https://bmcevolbiol.biomedcentral.com/articles/10.1186/1471-2148-8-64

---

## Round 0.3 · accepted · Accept

Your responses to the previous concerns have cleared up the issues that were outstanding and indeed made your findings stronger. I am happy with how the manuscript now stands and consider it ready for publication

#